# Mitochondrial Ca²⁺ uptake by the voltage-dependent anion channel 2 regulates cardiac rhythmicity

Hirohito Shimizu[1†], Johann Schredelseker[1†‡], Jie Huang[1†], Kui Lu[2†¶], Shamim Naghdi[3], Fei Lu[1], Sarah Franklin[4§], Hannah DG Fiji[2], Kevin Wang[1], Huanqi Zhu[2], Cheng Tian[1], Billy Lin[1], Haruko Nakano[1,5], Amy Ehrlich[3], Junichi Nakai[6], Adam Z Stieg[7,8], James K Gimzewski[2,7,8,9], Atsushi Nakano[1,5], Joshua I Goldhaber[10], Thomas M Vondriska[4], György Hajnóczky[3], Ohyun Kwon[2], Jau-Nian Chen[1*]

[1]Department of Molecular, Cell and Developmental Biology, University of California, Los Angeles, Los Angeles, United States; [2]Department of Chemistry and Biochemistry, University of California, Los Angeles, Los Angeles, United States; [3]MitoCare Center, Department of Pathology, Anatomy and Cell Biology, Thomas Jefferson University, Philadelphia, United States; [4]Department of Anesthesiology, University of California, Los Angeles, Los Angeles, United States; [5]Broad Center of Regenerative Medicine and Stem Cell Research, University of California, Los Angeles, Los Angeles, United States; [6]Brain Science Institute, Saitama University, Saitama, Japan; [7]California NanoSystems Institute, University of California, Los Angeles, Los Angeles, United States; [8]WPI Center for Materials Nanoarchitectonics, National Institute for Materials Science, Tsukuba, Japan; [9]School of Physics, Centre for Nanoscience and Quantum Information, University of Bristol, Bristol, UK; [10]Cedars-Sinai Heart Institute, Los Angeles, United States

**\*For correspondence:** chenjn@ mcdb.ucla.edu

[†]These authors contributed equally to this work

**Present address:** [‡]Walther-Straub Institute for Pharmacology and Toxicology, Ludwig-Maximilians University, Munich, Germany; [¶]College of Bioengineering, Tianjin University of Science and Technology, Tianjin, China; [§]Department of Internal Medicine, Nora Eccles Harrison Cardiovascular Research and Training Institute, University of Utah, Salt Lake City, United States

**Competing interests:** The authors declare that no competing interests exist.

**Abstract** Tightly regulated Ca²⁺ homeostasis is a prerequisite for proper cardiac function. To dissect the regulatory network of cardiac Ca²⁺ handling, we performed a chemical suppressor screen on zebrafish *tremblor* embryos, which suffer from Ca²⁺ extrusion defects. Efsevin was identified based on its potent activity to restore coordinated contractions in *tremblor*. We show that efsevin binds to VDAC2, potentiates mitochondrial Ca²⁺ uptake and accelerates the transfer of Ca²⁺ from intracellular stores into mitochondria. In cardiomyocytes, efsevin restricts the temporal and spatial boundaries of Ca²⁺ sparks and thereby inhibits Ca²⁺ overload-induced erratic Ca²⁺ waves and irregular contractions. We further show that overexpression of VDAC2 recapitulates the suppressive effect of efsevin on *tremblor* embryos whereas VDAC2 deficiency attenuates efsevin's rescue effect and that VDAC2 functions synergistically with MCU to suppress cardiac fibrillation in *tremblor*. Together, these findings demonstrate a critical modulatory role for VDAC2-dependent mitochondrial Ca²⁺ uptake in the regulation of cardiac rhythmicity.

## Introduction

During development, well-orchestrated cellular processes guide cells from diverse lineages to integrate into the primitive heart tube and establish rhythmic and coordinated contractions. While many genes and pathways important for cardiac morphogenesis have been identified, molecular

**eLife digest** The heart is a large muscle that pumps blood around the body by maintaining a regular rhythm of contraction and relaxation. If the heart loses this regular rhythm it works less efficiently, which can lead to life-threatening conditions.

Regular heart rhythms are maintained by changes in the concentration of calcium ions in the cytoplasm of the heart muscle cells. These changes are synchronised so that the heart cells contract in a controlled manner. In each cell, a contraction begins when calcium ions from outside the cell enter the cytoplasm by passing through a channel protein in the membrane that surrounds the cell. This triggers the release of even more calcium ions into the cytoplasm from stores within the cell. For the cells to relax, the calcium ions must then be pumped out of the cytoplasm to lower the calcium ion concentration back to the original level.

Shimizu et al. studied a zebrafish mutant—called *tremblor*—that has irregular heart rhythms because its heart muscle cells are unable to efficiently remove calcium ions from the cytoplasm. Embryos of the *tremblor* mutant were treated with a wide variety of chemical compounds with the aim of finding some that could correct the heart defect.

A compound called efsevin restores regular heart rhythms in *tremblor* mutants. Efsevin binds to a pump protein called VDAC2, which is found in compartments called mitochondria within the cell. Although mitochondria are best known for their role in supplying energy for the cell, they also act as internal stores for calcium. By binding to VDAC2, efsevin increases the rate at which calcium ions are pumped from the cytoplasm into the mitochondria. This restores rhythmic calcium ion cycling in the cytoplasm and enables the heart muscle cells to develop regular rhythms of contraction and relaxation. Increasing the levels of VDAC2 or another similar calcium ion pump protein in the heart cells can also restore a regular heart rhythm.

Efsevin can also correct irregular heart rhythms in human and mouse heart muscle cells, therefore the new role for mitochondria in controlling heart rhythms found by Shimizu et al. appears to be shared in other animals. The experiments have also identified the VDAC family of proteins as potential new targets for drug therapies to treat people with irregular heart rhythms.

mechanisms governing embryonic cardiac rhythmicity are poorly understood. The findings that $Ca^{2+}$ waves traveling across the heart soon after the formation of the primitive heart tube (*Chi et al., 2008*) and that loss of function of key $Ca^{2+}$ regulatory proteins, such as the L-type $Ca^{2+}$ channel, Na/K–ATPase and sodium-calcium exchanger 1 (NCX1), severely impairs normal cardiac function (*Rottbauer et al., 2001*; *Shu et al., 2003*; *Ebert et al., 2005*; *Langenbacher et al., 2005*), indicate an essential role for $Ca^{2+}$ handling in the regulation of embryonic cardiac function.

$Ca^{2+}$ homoeostasis in cardiac muscle cells is tightly regulated at the temporal and spatial level by a subcellular network involving multiple proteins, pathways, and organelles. The release and reuptake of $Ca^{2+}$ by the sarcoplasmic reticulum (SR), the largest $Ca^{2+}$ store in cardiomyocytes, constitutes the primary mechanism governing the contraction and relaxation of the heart. $Ca^{2+}$ influx after activation of the L-type $Ca^{2+}$ channel in the plasma membrane induces the release of $Ca^{2+}$ from the SR via ryanodine receptor (RyR) channels, which leads to an increase of the intracellular $Ca^{2+}$ concentration and cardiac contraction. During diastolic relaxation, $Ca^{2+}$ is transferred back into the SR by the SR $Ca^{2+}$ pump or extruded from the cell through NCX1. Defects in cardiac $Ca^{2+}$ handling and $Ca^{2+}$ overload, for example during cardiac ischemia/reperfusion or in long QT syndrome, are well known causes of contractile dysfunction and many types of arrhythmias including early and delayed after-depolarizations and Torsade des pointes (*Bers, 2002*; *Choi et al., 2002*; *Yano et al., 2008*; *Greiser et al., 2011*).

$Ca^{2+}$ crosstalk between mitochondria and ER/SR has been noted in many cell types and the voltage-dependent anion channel (VDAC) and the mitochondrial $Ca^{2+}$ uniporter (MCU) serve as primary routes for $Ca^{2+}$ entry through the outer and inner mitochondrial membranes, respectively (*Rapizzi et al., 2002*; *Bathori et al., 2006*; *Shoshan-Barmatz et al., 2010*; *Baughman et al., 2011*; *De Stefani et al., 2011*). In the heart, mitochondria are tethered to the SR and are located in close proximity to $Ca^{2+}$ release sites (*García-Pérez et al., 2008*; *Boncompagni et al., 2009*; *Hayashi et al., 2009*). This subcellular architecture exposes the mitochondria near the $Ca^{2+}$ release sites to a high local $Ca^{2+}$ concentration

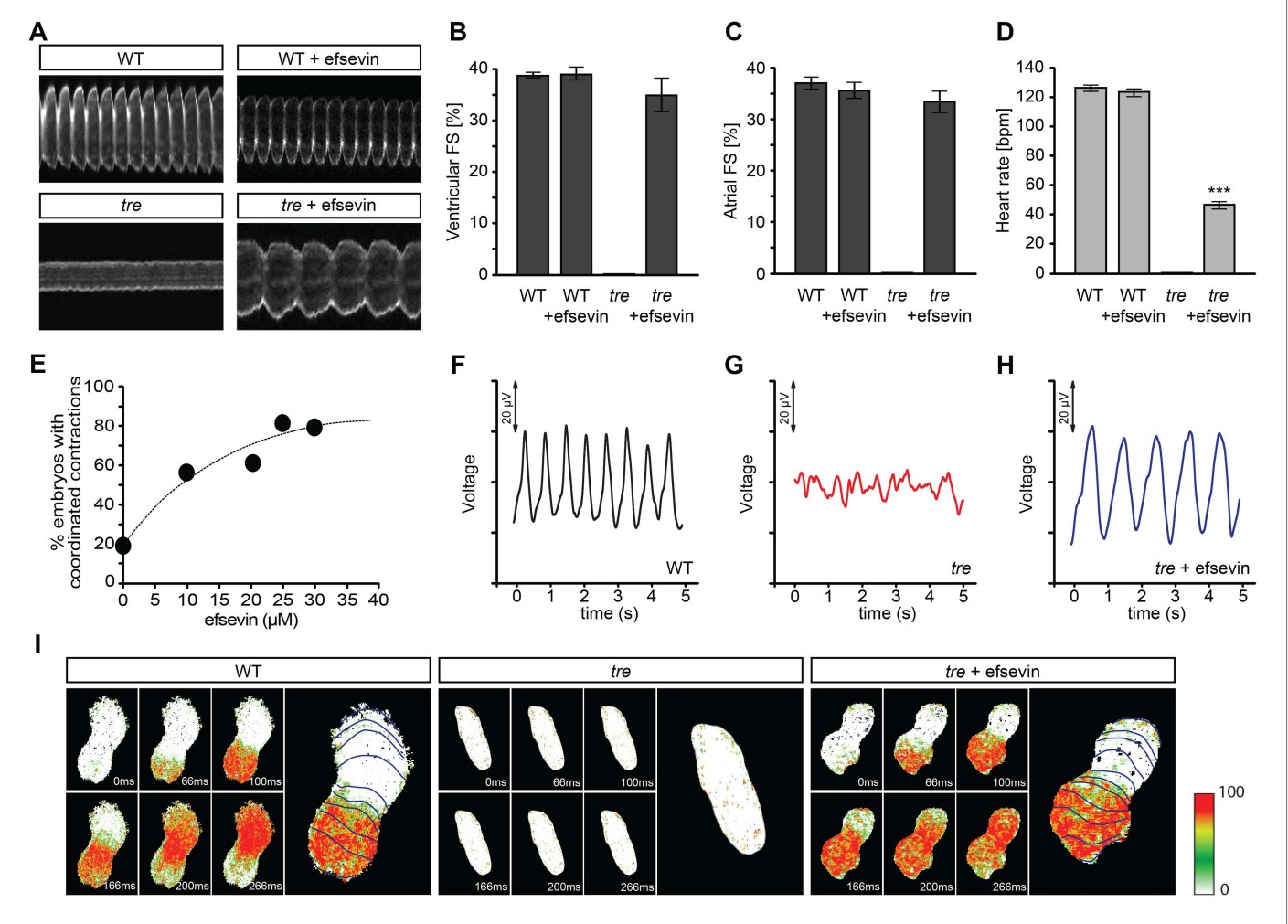

**Figure 1**. Efsevin restores rhythmic cardiac contractions in zebrafish tremblor embryos. (**A**) Line scans across the atria of *Tg(myl7:GFP)* embryonic hearts at 48 hpf. Rhythmically alternating systoles and diastoles are recorded from vehicle- (upper left) or efsevin- treated wild type (upper right) and efsevin-treated *tre* (lower right) embryos, while only sporadic unsynchronized contractions are recorded from vehicle-treated *tre* embryos (lower left). (**B**, **C**) Fractional shortening (FS) deduced from the line-scan traces. While cardiac contraction was not observed in *tre*, efsevin-treated wild type and *tre* hearts have similar levels of FS to those observed in control hearts. Ventricular FS of wild type v.s. wild type + efsevin vs tre + efsevin: 39 ± 0.6%, n = 8 vs 39 ± 1%, n = 10 vs 35 ± 3%, n = 6; and Atrial FS: 37 ± 1%, n = 11 vs 35 ± 2%, n = 11 vs 33 ± 2%, n = 15. (**D**) While efsevin restored a heart rate of 46 ± 2 beats per minute (bpm) in *tre* embryos, same treatment does not affect the heart rate in wild type embryos (126 ± 2 bpm in vehicle-treated embryos vs 123 ± 3 bpm in efsevin-treated wild-type embryos). ***, p < 0.001 by one-way ANOVA. (**E**) Dose-dependence curve for efsevin. The *tre* embryos were treated with various concentrations of efsevin from 24 hpf and cardiac contractions were analyzed at 48 hpf. (**F**–**H**) Representative time traces of local field potentials for wild type (**F**), *tre* (**G**) and efsevin-treated *tre* (**H**) embryos clearly display periods of regular, irregular, and restored periodic electrical activity. (**I**) In vivo optical maps of Ca$^{2+}$ activation represented by isochronal lines every 33 ms recorded from 36 hpf wild type (left), *tre* (center) and efsevin-treated *tre* (right) embryos.

that is sufficient to overcome the low Ca$^{2+}$ affinity of MCU and facilitates Ca$^{2+}$ crosstalk between SR and mitochondria (*García-Pérez et al., 2008*; *Dorn and Scorrano, 2010*; *Kohlhaas and Maack, 2013*). Increase of the mitochondrial Ca$^{2+}$ concentration enhances energy production during higher workload and dysregulation of SR-mitochondrial Ca$^{2+}$ signaling results in energetic deficits and oxidative stress in the heart and may trigger programmed cell death (*Brandes and Bers, 1997*; *Maack et al., 2006*; *Kohlhaas and Maack, 2013*). However, whether SR-mitochondrial Ca$^{2+}$ crosstalk also contributes significantly to cardiac Ca$^{2+}$ signaling during excitation-contraction coupling requires further investigation.

In zebrafish, the *tremblor* (*tre*) locus encodes a cardiac-specific isoform of the Na$^{+}$/Ca$^{2+}$ exchanger 1, NCX1h (also known as slc8a1a) (*Ebert et al., 2005*; *Langenbacher et al., 2005*). The *tre* mutant hearts

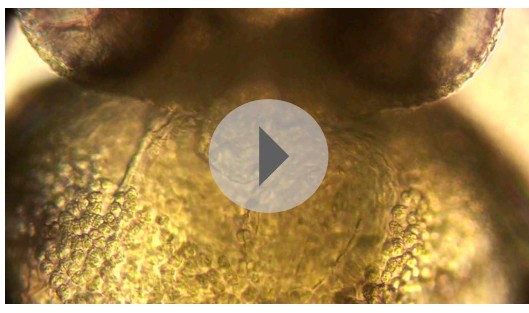

**Video 1**. The video shows a heart of a wild-type zebrafish embryo at 2 dpf. Robust rhythmic contractions can be observed in atrium and ventricle.

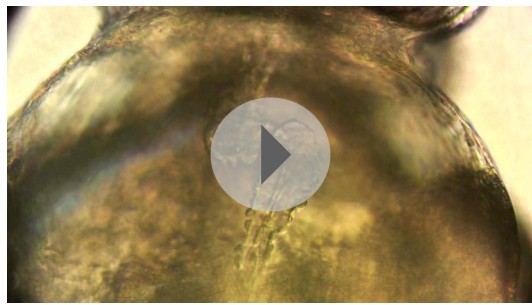

**Video 2**. This video shows a heart of a *tremblor* embryo at 2 dpf. Embryos of the mutant line *tremblor* display only local, unsynchronized contractions, comparable to cardiac fibrillation.

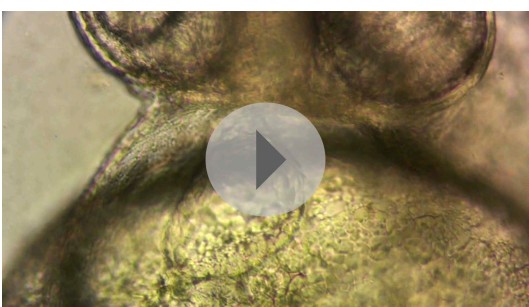

**Video 3**. This video shows a heart of a *tremblor* embryo at 2 dpf treated with efsevin. Treatment of *tremblor* embryos with efsevin restores rhythmic contractions with comparable atrial fractional shortening compared to wild-type embryos and approximately 40% of wild-type heart rate.

lack rhythmic $Ca^{2+}$ transients and display chaotic $Ca^{2+}$ signals in the myocardium leading to unsynchronized contractions resembling cardiac fibrillation (*Langenbacher et al., 2005*). In this study, we used *tre* as an animal model for aberrant $Ca^{2+}$ handling-induced cardiac dysfunction and took a chemical genetic approach to dissect the $Ca^{2+}$ regulatory network important for maintaining cardiac rhythmicity. A synthetic compound named efsevin was identified from a suppressor screen due to its potent ability to restore coordinated contractions in *tre*. Using biochemical and genetic approaches we show that efsevin interacts with VDAC2 and potentiates its mitochondrial $Ca^{2+}$ transporting activity and spatially and temporally modulates cytosolic $Ca^{2+}$ signals in cardiomyocytes. The important role of mitochondrial $Ca^{2+}$ uptake in regulating cardiac rhythmicity is further supported by the suppressive effect of VDAC2 and MCU overexpression on cardiac fibrillation in *tre*.

## Results and discussion

### Identification of a chemical suppressor of *tre* cardiac dysfunction

Homozygous *tre* mutant embryos suffer from $Ca^{2+}$ extrusion defects and manifest chaotic cardiac contractions resembling fibrillation (*Ebert et al., 2005*; *Langenbacher et al., 2005*). To dissect the regulatory network of $Ca^{2+}$ handling in cardiomyocytes and to identify mechanisms controlling embryonic cardiac rhythmicity, we screened the BioMol library and a collection of synthetic compounds for chemicals that are capable of restoring heartbeat either completely or partially in *tre* embryos. A dihydropyrrole carboxylic ester compound named efsevin was identified based on its ability to restore persistent and rhythmic cardiac contractions in *tre* mutant embryos in a dose-dependent manner (*Figure 1A,E*, and *Videos 1–4*). To validate the effect of efsevin, we assessed cardiac performance of wild type, *tre* and efsevin-treated *tre* embryos (*Nguyen et al., 2009*). Fractional shortening of efsevin treated *tre* mutant hearts was comparable to that of their wild type siblings and heart rate was restored to approximately 40% of that observed in controls (*Figure 1B–D*). Periodic local field potentials accompanying each heartbeat were detected in wild type and efsevin-treated *tre* embryos using a microelectrode array (*Figure 1F–H*). Furthermore, while only sporadic $Ca^{2+}$ signals were detected in *tre* hearts, in vivo $Ca^{2+}$ imaging revealed steady $Ca^{2+}$ waves propagating through efsevin-treated *tre* hearts (*Figure 1I*, *Videos 5–7*), demonstrating that cardiomyocytes are functionally coupled and that efsevin treatment restores regular $Ca^{2+}$ transients in *tre* hearts.

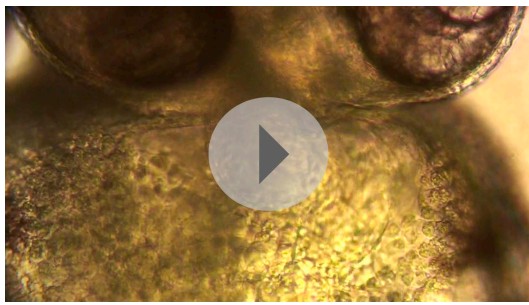

**Video 4**. The video shows a heart of a wild-type zebrafish embryo at 2 dpf treated with efsevin. Treatment of wild-type embryos with efsevin did not affect cardiac performance, indicated by robust, rhythmic contractions comparable to untreated wild-type embryos.

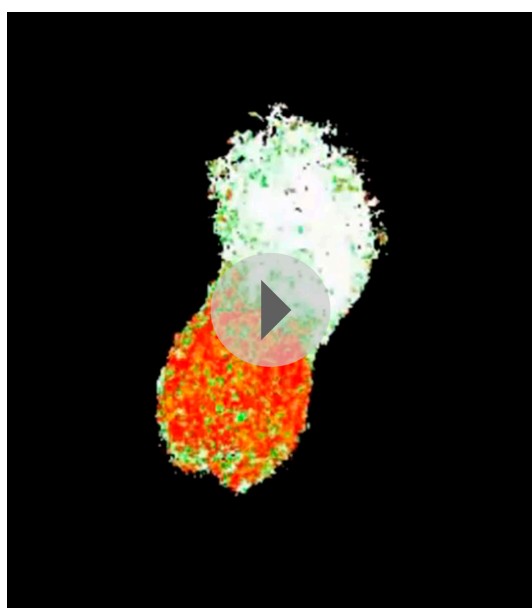

**Video 5**. Heat map of Ca²⁺ transients recorded in 1 day old wild type heart.

## Efsevin suppresses Ca²⁺ overload-induced irregular contraction

We next examined whether efsevin could suppress aberrant Ca²⁺ homeostasis-induced arrhythmic responses in mammalian cardiomyocytes. Mouse embryonic stem cell-derived cardiomyocytes (mESC-CMs) establish a regular contraction pattern with rhythmic Ca²⁺ transients (*Figure 2A,B,E,F*). Mimicking Ca²⁺ overload by increasing extracellular Ca²⁺ levels was sufficient to disrupt normal Ca²⁺ cycling and induce irregular contractions in mESC-CMs (*Figure 2C,E,F*). Remarkably, efsevin treatment restored rhythmic Ca²⁺ transients and cardiac contractions in these cells (*Figure 2D–F*). Similar effect was observed in human embryonic stem cell-derived cardiomyocytes (hESC-CMs) (*Figure 2G*). Together, these findings suggest that efsevin targets a conserved Ca²⁺ regulatory mechanism critical for maintaining rhythmic cardiac contraction in fish, mice and humans.

## VDAC2 mediates the suppressive effect of efsevin on *tre*

To identify the protein target of efsevin, we generated a N-Boc-protected 2-aminoethoxyethoxyethylamine linker-attached efsevin (efsevinᴸ) (*Figure 3A,C*). This modified compound retained the activity of efsevin to restore cardiac contractions in *ncx1h* deficient embryos (*Figure 3B,D*) and was used to create efsevin-conjugated agarose beads (efsevinᴸᴮ). A 32kD protein species was detected from zebrafish lysate due to its binding ability to efsevinᴸᴮ and OK-C125ᴸᴮ, an active efsevin derivative conjugated to beads, but not to beads capped with ethanolamine alone or beads conjugated with an inactive efsevin analog (OK-C19ᴸᴮ) (*Figure 3A–E*). Furthermore, preincubation of zebrafish lysate with excess efsevin prevented the 32kD protein from binding to efsevinᴸᴮ or OK-C125ᴸᴮ (*Figure 3E*). Mass spectrometry analysis revealed that this 32kD band represents a zebrafish homologue of the mitochondrial voltage-dependent anion channel 2 (VDAC2) (*Figure 3F* and *Figure 3—figure supplement 1*).

VDAC2 is expressed in the developing zebrafish heart (*Figure 4A*), making it a good candidate for mediating efsevin's effect on cardiac Ca²⁺ handling. To examine this possibility, we injected in vitro synthesized VDAC2 RNA into *tre* embryos and found that the majority of these embryos had coordinated cardiac contractions similar to those subjected to efsevin treatment (*Figure 4B*, *Videos 8–11*). In addition, we generated *myl7:VDAC2* transgenic fish in which VDAC2 expression can be induced in the heart by tebufenozide (TBF) (*Figure 4C*). Knocking down NCX1h in *myl7:VDAC2* embryos results in chaotic cardiac movement similar to *tre*. Like efsevin treatment, induction of VDAC2 expression by TBF treatment restored coordinated and rhythmic contractions in *myl7:VDAC2;NCX1h MO* hearts (*Figure 4D*, *Videos 12,13*). Conversely, knocking down VDAC2 in *tre* hearts attenuated the suppressive effect of efsevin (*Figure 4E*, *Videos 14–16*). Furthermore, we generated VDAC2 null embryos by the Zinc Finger Nuclease gene targeting approach (*Figure 4G*). Similar to that observed in morpholino knockdown embryos, homozygous *VDAC2^LA2256* embryos do not exhibit noticeable morphological

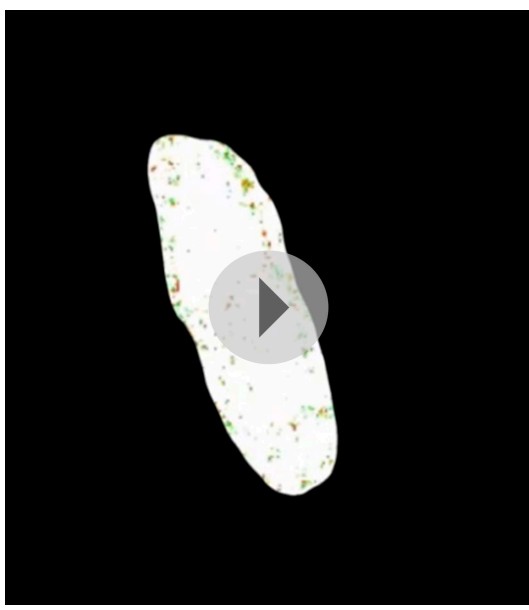

**Video 6**. Heat map of Ca²⁺ transients recorded in 1 day old *tremblor* heart.

**Video 7**. Heat map of Ca²⁺ transients recorded in 1 day old efsevin treated *tremblor* heart.

defects, but the suppressive effect of efsevin was attenuated in homozygous *VDAC2*[LA2256]; *NCX1MO* embryos (*Figure 4F*). These findings demonstrate that VDAC2 is a major mediator for efsevin's effect on *ncx1h* deficient hearts.

## VDAC2-dependent effect of efsevin on mitochondrial Ca²⁺ uptake

VDAC is an abundant channel located on the outer mitochondrial membrane serving as a primary passageway for metabolites and ions (*Figure 5A*) (*Rapizzi et al., 2002*; *Bathori et al., 2006*; *Shoshan-Barmatz et al., 2010*). At its close state, VDAC favours Ca²⁺ flux (*Tan and Colombini, 2007*). To examine whether efsevin would modulate mitochondrial Ca²⁺ uptake via VDAC2, we transfected HeLa cells with VDAC2. We noted increased mitochondrial Ca²⁺ uptake in permeabilized VDAC2 transfected and efsevin-treated cells after the addition of Ca²⁺ and the combined treatment further enhanced mitochondrial Ca²⁺ levels (*Figure 5B*).

Mitochondria are located in close proximity to Ca²⁺ release sites of the ER/SR and an extensive crosstalk between the two organelles exists (*García-Pérez et al., 2008*; *Hayashi et al., 2009*; *Brown and O'Rourke, 2010*; *Dorn and Scorrano, 2010*; *Kohlhaas and Maack, 2013*). We examined whether Ca²⁺ released from intracellular stores could be locally transported into mitochondria through VDAC2 in VDAC1/VDAC3 double knockout (V1/V3DKO) MEFs where VDAC2 is the only VDAC isoform being expressed (*Roy et al., 2009a*). While treatments with ATP, an IP3-linked agonist, and thapsigargin, a SERCA inhibitor, stimulated similar global cytoplasmic [Ca²⁺] elevation in intact cells, only ATP induced a rapid mitochondrial matrix [Ca²⁺] rise (*Figure 5—figure supplement 1*). This finding is consistent with observations obtained in other cell types (*Rizzuto et al., 1994*; *Hajnóczky et al., 1995*) and suggests that Ca²⁺ was locally transferred from IP3 receptors to mitochondria through VDAC2 at the close ER-mitochondrial associations. We next investigated whether this process could be modulated by efsevin. In permeabilized V1/V3DKO MEFs, treatment with efsevin increased the amount of Ca²⁺ transferred into mitochondria during IP₃-induced Ca²⁺ release (*Figure 5C*). Also, in intact V1/V3 DKO MEFs, efsevin accelerated the transfer of Ca²⁺ released from intracellular stores into mitochondria during stimulation with ATP (*Figure 5D,E*).

## Efsevin modulates Ca²⁺ sparks and suppresses erratic Ca²⁺ waves in cardiomyocytes

We next examined the effect of efsevin on cytosolic Ca²⁺ signals in isolated adult murine cardiomyocytes. We found that efsevin treatment induced faster inactivation kinetics without affecting the

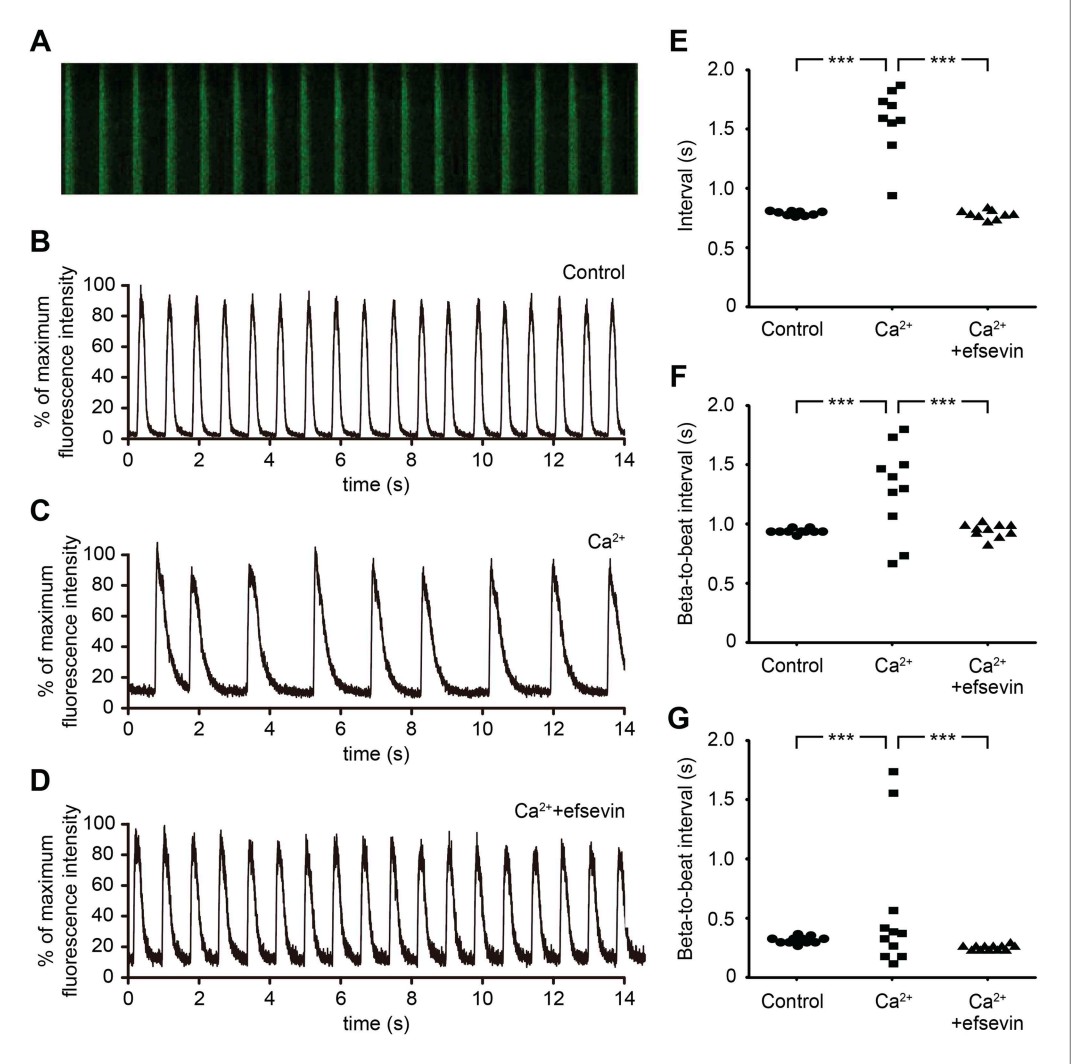

**Figure 2**. Efsevin reduces arrhythmogenic events in ES cell-derived cardiomyocytes. (**A**) Line-scan analysis of Ca²⁺ transients in mESC-CMs after 10 days of differentiation. (**B–D**) Representative graph of Ca²⁺ transients detected in mESC-CMs (**B**). After treatment with 10 mM Ca²⁺ for 10 min, the EB showed an irregular pattern of Ca²⁺ transients (**C**). Efsevin treatment restores regular Ca²⁺ transients under Ca²⁺ overload conditions in mESC-CMs (**D**). (**E**) Plotted intervals between peaks of Ca²⁺ signals detected in mESC-CMs prior to treatment (control), in 10 mM Ca²⁺$_{ext}$ (Ca²⁺) and in 10 mM Ca²⁺$_{ext}$+10 µM efsevin (Ca²⁺+efsevin). (**F, G**) Plotted intervals of contractions detected in EBs prior to treatment (control), in 10 mM Ca²⁺$_{ext}$ (Ca²⁺) and in 10 mM Ca²⁺$_{ext}$ + 10 µM efsevin (Ca²⁺ + efsevin) for mouse ESC-CMs (**F**) and 5 mM Ca²⁺$_{ext}$ (Ca²⁺) and in 5 mM Ca²⁺$_{ext}$ + 5 µM efsevin (Ca²⁺ + efsevin) for human ESC-CMs (**G**). ***, p < 0.001 by F-test.

amplitude or time to peak of paced Ca²⁺ transients (*Figure 6A*). Similarly, efsevin treatment did not significantly alter the frequency, amplitude or Ca²⁺ release flux of spontaneous Ca²⁺ sparks, local Ca²⁺ release events, but accelerated the decay phase resulting in sparks with a shorter duration and a narrower width (*Figure 6B*). These results indicate that by activating mitochondrial Ca²⁺ uptake, efsevin accelerates Ca²⁺ removal from the cytosol in cardiomyocytes and thereby restricts local cytosolic Ca²⁺ sparks to a narrower domain for a shorter period of time without affecting SR Ca²⁺ load or RyR Ca²⁺ release. Under conditions of Ca²⁺ overload, single Ca²⁺ sparks can trigger opening of neighbouring Ca²⁺ release units and thus induce the formation of erratic Ca²⁺ waves (*Figure 6C*). Efsevin treatment significantly reduced the number of propagating Ca²⁺ waves in a dosage-dependent manner (*Figure 6C,D*), demonstrating a potent suppressive effect of efsevin on the propagation of Ca²⁺ overload-induced Ca²⁺ waves and suggesting that efsevin could serve as a pharmacological tool to manipulate local Ca²⁺ signals.

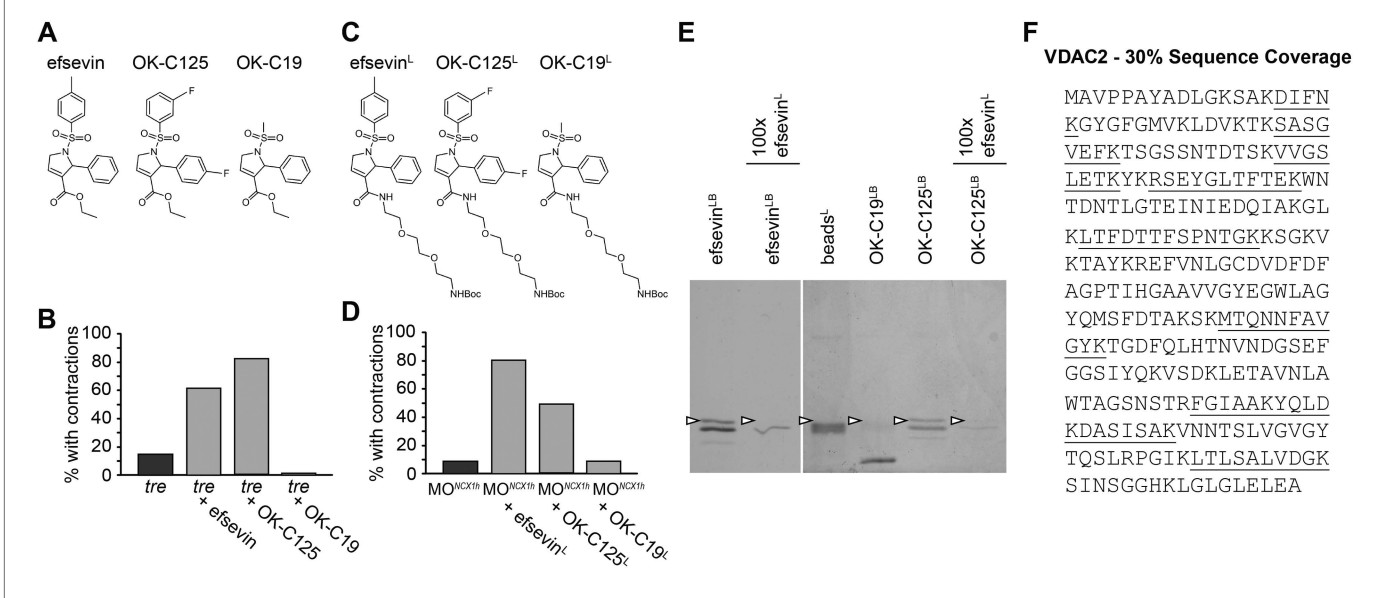

**Figure 3**. VDAC2 is a protein target of efsevin. (**A**) Structures of efsevin and two derivatives, OK-C125 and OK-C19. (**B**) Efsevin and OK-C125 restored rhythmic contractions in the majority of *tremblor* embryos, whereas OK-C19 failed to rescue the *tremblor* phenotype. (**C**) Structures of linker-attached compounds (indicated by superscript L). (**D**) Compounds efsevin[L] and OK-C125[L] retained their ability to restore rhythmic contractions in NCX1hMO injected embryos, while the inactive derivative OK-C19[L] was still unable to induce rhythmic contraction. (**E**) Affinity agarose beads covalently linked with efsevin (efsevin[LB]) or OK-C125 (OK-C125[LB]) pulled down 2 protein species from zebrafish embryonic lysate, whereof one, the 32 kD upper band, was sensitive to competition with a 100-fold excess free efsevin[L]. The 32 kD band was not detected in proteins eluted from beads capped with ethanolamine alone (beads[C]) or beads linked to OK-C19 (OK-C19[LB]). Arrowheads point to the 32kD bands. (**F**) Mass Spectrometry identifies the 32kD band as VDAC2. Peptides identified by mass spectrometry (underlined) account for 30% of the total sequence.

The following figure supplement is available for figure 3:

**Figure supplement 1**. Mass Spectometry identifies VDAC2 as the target of efsevin.

## Mitochondrial Ca²⁺ uptake modulates embryonic cardiac rhythmicity

We hypothesize that efsevin treatment/VDAC2 overexpression suppresses aberrant $Ca^{2+}$ handling-associated arrhythmic cardiac contractions by buffering excess $Ca^{2+}$ into mitochondria. This hypothesis predicts that activating other mitochondrial $Ca^{2+}$ uptake molecules would likewise restore coordinated contractions in *tre*. To test this model, we cloned zebrafish MCU and MICU1, an inner mitochondrial membrane $Ca^{2+}$ transporter and its regulator (*Perocchi et al., 2010*; *Baughman et al., 2011*; *De Stefani et al., 2011*; *Mallilankaraman et al., 2012*; *Csordas et al., 2013*). In situ hybridization showed that MCU and MICU1 were expressed in the developing zebrafish heart (*Figure 7A*) and their expression levels were comparable between the wild type and *tre* hearts (*Figure 7—figure supplement 1*). Overexpression of MCU restored coordinated contractions in *tre*, akin to what was observed with VDAC2 (*Figure 7B*). In addition, *tre* embryos injected with suboptimal concentrations of MCU or VDAC2 had a fibrillating heart, but embryos receiving both VDAC2 and MCU at the suboptimal concentration manifested coordinated contractions (*Figure 7C*), demonstrating a synergistic effect of these proteins. Furthermore, overexpression of MCU failed to suppress the *tre* phenotype in the absence of VDAC2 activity and VDAC2 could not restore coordinated contractions in *tre* without functional MCU (*Figure 7B,D*). Similar results were observed by manipulating MICU1 activity (*Figure 7E,F*). Together, these findings indicate that mitochondrial $Ca^{2+}$ uptake mechanisms on outer and inner mitochondrial membranes act cooperatively to regulate cardiac rhythmicity.

## Conclusion

In summary, we conducted a chemical suppressor screen in zebrafish to dissect the regulatory network critical for maintaining rhythmic cardiac contractions and to identify mechanisms underlying aberrant

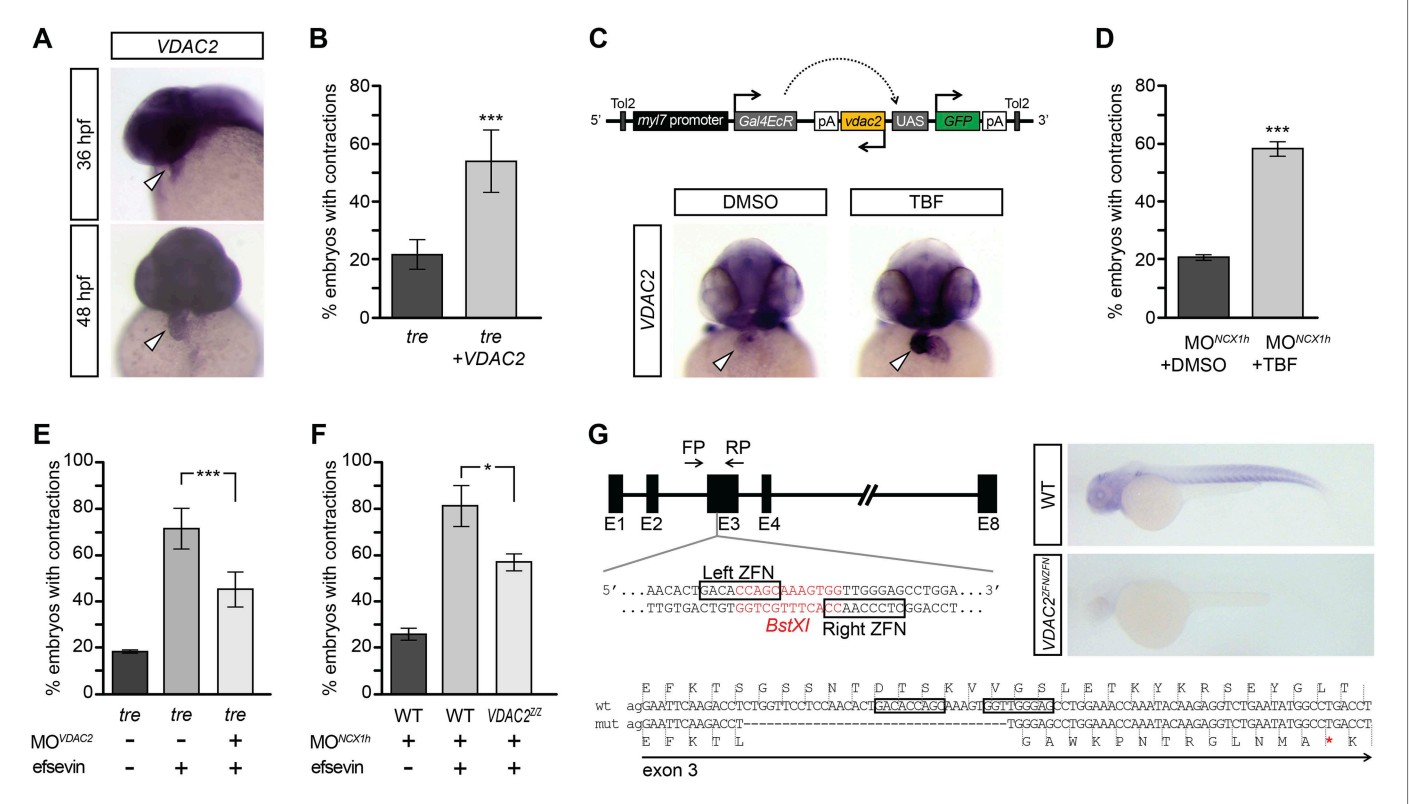

**Figure 4**. VDAC2 restores rhythmic cardiac contractions in *tre*. (**A**) In situ hybridization analysis showed that VDAC2 is expressed in embryonic hearts at 36 hpf (upper image) and 48 hpf (lower image). (**B**) Injection of 25 pg in vitro synthesized VDAC2 mRNA restored cardiac contractions in 52.9 ± 12.1% (n = 78) of 1-day-old *tre* embryos, compared to 21.8 ± 5.1% in uninjected siblings (n = 111). (**C**) Schematic diagram of *myl7:VDAC2* construct (top). In situ hybridization analysis showed that TBF treatment induces VDAC2 expression in the heart (lower panel). (**D**) While only ~20% of *myl7:VDAC2;NCX1hMO* embryos have coordinated contractions (n = 116), 52.3 ± 2.4% of these embryos established persistent, rhythmic contractions after TBF induction of VDAC2 (n = 154). (**E**) On average, 71.2 ± 8.8% efsevin treated embryos have coordinated cardiac contractions (n = 131). Morpholino antisense oligonu-cleotide knockdown of VDAC2 (MO$^{VDAC2}$) attenuates the ability of efsevin to suppress cardiac fibrillation in *tre* embryos (45.3 ± 7.4% embryos with coordinated contractions, n = 94). (**F**) Efsevin treatment restores coordinated cardiac contractions in 76.2 ± 8.7% NCX1MO embryos, only 54.1 ± 3.6% VDAC2$^{zfn/zfn}$;NCX1MO embryos have coordinated contractions (n = 250). (**G**) Diagram of Zinc finger target sites. *VDAC2$^{zfn/zfn}$* carries a 34 bp deletion in exon 3 which results in a premature stop codon (red asterisk). In situ hybridization analysis showing loss of VDAC2 transcripts in *VDAC2$^{zfn/zfn}$* embryos. White arrowheads point to the developing heart.

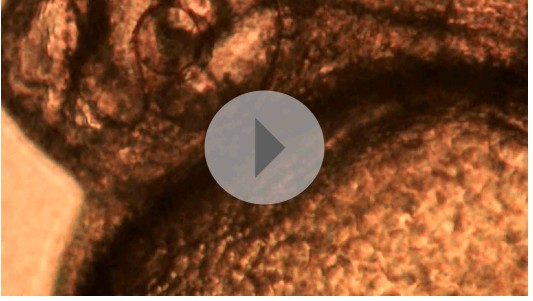

**Video 8**. This video shows a heart of a wild-type zebrafish embryo at 1 dpf. Robust rhythmic contractions can be observed in atrium and ventricle.

Ca$^{2+}$ handling-induced cardiac dysfunction. We show that activation of VDAC2 through overex-pression or efsevin treatment potently restores rhythmic contractions in NCX1h deficient zebrafish hearts and effectively suppresses Ca$^{2+}$ over-load-induced arrhythmogenic Ca$^{2+}$ events and irregular contractions in mouse and human cardi-omyocytes. We provide evidence that potentiat-ing VDAC2 activity enhances mitochondrial Ca$^{2+}$ uptake, accelerates Ca$^{2+}$ transfer from intracel-lular stores into mitochondria and spatially and temporally restricts single Ca$^{2+}$ sparks in cardio-myocytes. The crucial role of mitochondria in the regulation of cardiac rhythmicity is further sup-ported by the findings that VDAC2 functions in concert with MCU; these genes have a strong synergistic effect on suppressing cardiac fibrillation and loss of function of either gene abrogates the rescue effect of the other in *tre*.

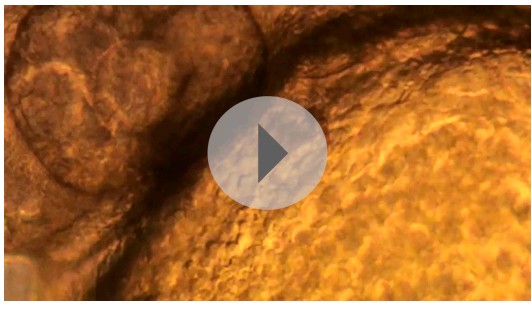

**Video 9**. This video shows a heart of a wild-type zebrafish embryo injected with zebrafish VDAC2 mRNA at 1 dpf. Robust rhythmic contractions can be observed in atrium and ventricle.

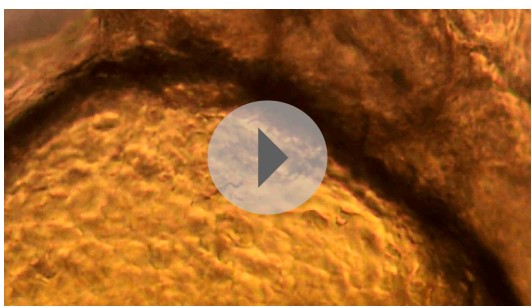

**Video 10**. This video shows a heart of a *tremblor* embryo at 1 dpf. *Tremblor* embryos display only local, unsynchronized contractions, comparable to cardiac fibrillation.

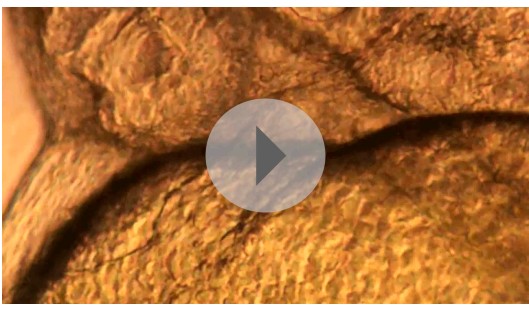

**Video 11**. This video shows a heart of a *tremblor* embryo injected with zebrafish VDAC2 mRNA at 1 dpf. Overexpression of zebrafish VDAC2 mRNA restores rhythmic contractions in *tremblor* embryos.

The regulatory roles of mitochondrial $Ca^{2+}$ in cardiac metabolism, cell survival and fate have been studied extensively (*Brown and O'Rourke, 2010*; *Dorn and Scorrano, 2010*; *Doenst et al., 2013*; *Kasahara et al., 2013*; *Kohlhaas and Maack, 2013*; *Luo and Anderson, 2013*). Our study provides genetic and physiologic evidence supporting an additional role for mitochondria in regulating cardiac rhythmicity and reveals VDAC2 as a modulator of $Ca^{2+}$ handling in cardiomyocytes. Our findings, together with recent reports of the physical interaction between VDAC2 and RyR2 (*Min et al., 2012*) and the close proximity of outer and inner mitochondrial membranes at the contact sites between the mitochondria and the SR (*García-Pérez et al., 2011*), suggest an intriguing model. We propose that mitochondria facilitate an efficient clearance mechanism in the $Ca^{2+}$ microdomain, which modulates $Ca^{2+}$ handling without affecting global $Ca^{2+}$ signals in cardiomyocytes. In this model, VDAC facilitates mitochondrial $Ca^{2+}$ uptake via MCU complex and thereby controls the duration and the diffusion of cytosolic $Ca^{2+}$ near the $Ca^{2+}$ release sites to ensure rhythmic cardiac contractions. This model is consistent with our observation that efsevin treatment induces faster inactivation kinetics of cytosolic $Ca^{2+}$ transients without affecting the amplitude or the time to peak in cardiomyocytes and the reports that blocking mitochondrial $Ca^{2+}$ uptake has little impact on cytosolic $Ca^{2+}$ transients (*Maack et al., 2006*; *Kohlhaas et al., 2010*). Further support for this model comes from the observation of the $Ca^{2+}$ peaks on the OMM (*Drago et al., 2012*) and the finding that downregulating VDAC2 extends $Ca^{2+}$ sparks (*Subedi et al., 2011*; *Min et al., 2012*) and that blocking mitochondrial $Ca^{2+}$ uptake by Ru360 leads to an increased number of spontaneous propagating $Ca^{2+}$ waves (*Seguchi et al., 2005*). Future studies on the kinetics of VDAC2-dependent mitochondrial $Ca^{2+}$ uptake and exploring potential regulatory molecules for VDAC2 activity will provide insights into how the crosstalk between SR and mitochondria contributes to $Ca^{2+}$ handling and cardiac rhythmicity.

Aberrant $Ca^{2+}$ handling is associated with many cardiac dysfunctions including arrhythmia. Establishing animal models to study molecular mechanisms and develop new therapeutic strategies are therefore major preclinical needs. Our chemical suppressor screen identified a potent effect of efsevin and its biological target VDAC2 on manipulating cardiac $Ca^{2+}$ handling and restoring regular cardiac contractions in fish and mouse and human cardiomyocytes. This success indicates that fundamental mechanisms regulating cardiac function are conserved among vertebrates despite the existence of species-specific features and suggests a new paradigm of using zebrafish cardiac disease models for the dissection of critical genetic pathways and the discovery of new therapeutic approaches. Future studies examining the effects of efsevin on other

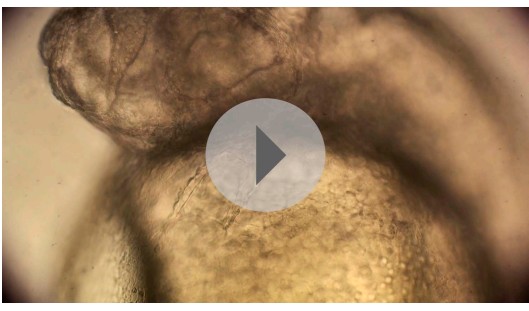

**Video 12**. This video shows a heart of a 2 dpf Tg-VDAC2 embryo injected with a morpholino targeting NCX1h. Morpholino knock-down of NCX1h results in a fibrillating heart.

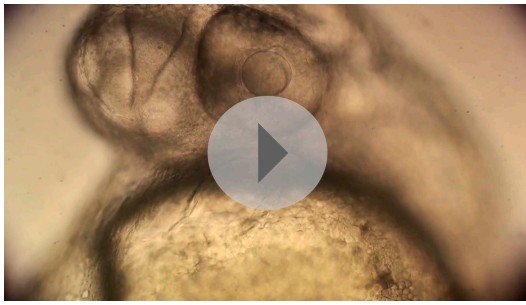

**Video 13**. This video shows a heart of a 2 dpf NCX1h morphant in the Tg-VDAC2 genetic background. TBF treatment induces VDAC2 expression and restores coordinated cardiac contractions.

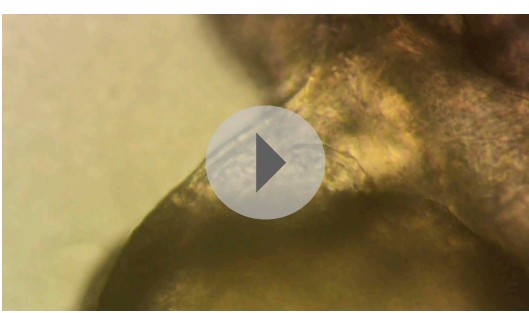

**Video 14**. This video shows a heart of a 2 dpf wild type zebrafish embryo injected with a morpholino targeting VDAC2. Morpholino knockdown of VDAC2 did not have obvious effects on cardiac performance.

arrhythmia models would further elucidate the potential for efsevin as a pharmacological tool to treat cardiac arrhythmia associated with aberrant $Ca^{2+}$ handling.

## Materials and methods

### Zebrafish husbandry and transgenic lines

Zebrafish of the mutant line *tremblor* (*tre*[tc318]) were maintained and bred as described previously (*Langenbacher et al., 2005*). Transgenic lines, *myl7:gCaMP4.1*[LA2124] and *myl7:VDAC2*[LA2309] were created using the Tol2kit (*Esengil et al., 2007*; *Kwan et al., 2007*; *Shindo et al., 2010*). The *VDAC2*[LA2256] was created using the zinc finger array OZ523 and OZ524 generated by the zebrafish Zinc Finger Consortium (*Foley et al., 2009a*, *2009b*).

### Molecular Biology

Full length VDAC2 cDNA was purchased from Open Biosystems (Huntsville, AL) and cloned into pCS2+ or pCS2+3XFLAG. Full length cDNA fragments of zebrafish MCU (Accession number: JX424822) and MICU1 (JX42823) were amplified from 2 dpf embryos and cloned into pCS2+. For mRNA synthesis, plasmids were linearized and mRNA was synthesized using the SP6 mMES-SAGE mMachine kit according to the manufacturers manual (Ambion, Austin, TX.).

### Zebrafish injections

VDAC2 mRNA and morpholino antisense oligos (5'-GGGAACGGCCATTTTATCTGTTAAA-3') (Genetools, Philomath, OR) were injected into one-cell stage embryos collected from crosses of *tre*[tc318] heterozygotes. Cardiac performance was analyzed by visual inspection on 1 dpf. The *tre* mutant embryos were identified either by observing the fibrillation phenotype at 2–3 dpf or by genotyping as previously described (*Langenbacher et al., 2005*).

### Chemical screen

Chemicals from a synthetic library (*Castellano et al., 2007*; *Choi et al., 2011*; *Cruz et al., 2011*) and from Biomol International LP (Farmingdale, NY) were screened for their ability to partially or completely restore persistent heartbeat in *tre* embryos. 12 embryos collected from crosses of

*tre*[tc318] heterozygotes were raised in the presence of individual compounds at a concentration of 10 μM from 4 hpf (*Choi et al., 2011*). Cardiac function was analyzed by visual inspection at 1 and 2 dpf. The hearts of *tre*[tc318] embryos manifest a chaotic movement resembling cardiac fibrillation with intermittent contractions in rare occasion (*Ebert et al., 2005*; *Langenbacher et al., 2005*). Compounds that elicit persistent coordinated cardiac contractions were validated on large number of *tre* mutant embryos and NCX1h morphants (>500 embryos).

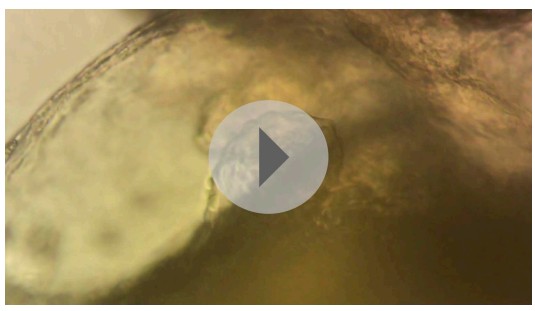

**Video 15**. This video shows a heart of a 2 dpf *tremblor* mutant embryo injected with a morpholino targeting VDAC2.

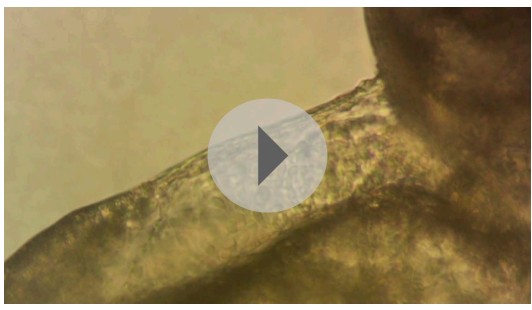

**Video 16**. This video shows a heart of a 2 dpf *tremblor* mutant embryo injected with a morpholino targeting VDAC2. Efsevin treatment cannot restore coordinated cardiac contractions in the absence of VDAC2.

## Zebrafish cardiac imaging

Videos of GFP-labelled *myl7:GFP* hearts were taken at 30 frames per second. Line-scan analysis was performed along a line through the atria or the ventricles of these hearts (*Nguyen et al., 2009*). Fraction of shortening was deduced from the ratio of diastolic and systolic width and heart rate was determined by beats per minute. Cardiac parameters were analyzed in *tremblor^tc318* and *VDAC2^LA2256* at 2 dpf.

## Zebrafish optical mapping

36 hpf *myl7:gCaMP4.1* embryos were imaged at a frame rate of 30 ms/frame. Electromechanical isolation was achieved by tnnt2MO (*Milan et al., 2006*). The fluorescence intensity of each pixel in a 2D map was normalize to generate heat maps and isochronal lines at 33 ms intervals were obtained by identifying the maximal spatial gradient for a given time point (*Chi et al., 2008*).

## Mouse and human embryonic stem cells

The mouse E14Tg2a ESC and human H9 ESC line were cultured and differentiated as previously described (*Blin et al., 2010*; *Arshi et al., 2013*). At day 10 of differentiation, beating mouse EBs were exposed to external solution containing 10 mM CaCl$_2$ for 10 min before DMSO or efsevin (10 µM) treatment. Human EBs were differentiated for 15 days and treated with 5 mM CaCl$_2$ for 10 min before DMSO or efsevin (5 µM) treatment. Images of beating EBs were acquired at a rate of 30 frames/s and analyzed by motion-detection software. For calcium recording, the EBs were loaded with 10 µM fluo-4 AM in culture media for 30 min at 37°C. Line-scan analysis was performed and fluorescent signals were acquired by a Zeiss LSM510 confocal microscope.

## Microelectrode array measurements

2-day-old wild type, *tre*, and efsevin-treated *tre* embryos were placed on uncoated, microelectrode arrays (MEAs) containing 120 integrated TiN electrodes (30 µm diameter, 200 µm interelectrode spacing). Local field potentials (LFPs) at each electrode were collected for three trials per embryo type over a period of three minutes at a sampling rate of 1 kHz using the MEA2100-HS120 system (Multichannel Systems, Reutiligen, Germany). Raw data was low-pass filtered at a cutoff frequency of 10 Hz using a third-order Butterworth filter. Data analysis was carried out using the MC_DataTool (Multichannel Systems) and Matlab (MathWorks).

## Ca$^{2+}$ imaging

Murine ventricular cardiomyocytes were isolated as previously described (*Reuter et al., 2004*). Cells were loaded with 5 µM fluo-4 AM in external solution containing: 138.2 mM NaCl, 4.6 mM KCl, 1.2 mM MgCl, 15 mM glucose, 20 mM HEPES for 1 hr and imaged in external solution supplemented with 2, 5 or 10 mM CaCl$_2$. For the recording of Ca$^{2+}$ sparks and transients, the external solution contained 2 mM CaCl$_2$. For Ca$^{2+}$ transients, cells were field stimulated at 0.5 Hz with a 5 ms pulse at a voltage of 20% above contraction threshold. For all measurements, efsevin was added 2 hr prior to the actual experiment. Images were recorded on a Zeiss LSM 5 Pascal confocal microscope. Data analysis was carried out using the Zeiss LSM Image Browser and ImageJ with the SparkMaster plugin (*Picht et al., 2007*). Cells were visually inspected prior to and after each recording. Only those recordings from

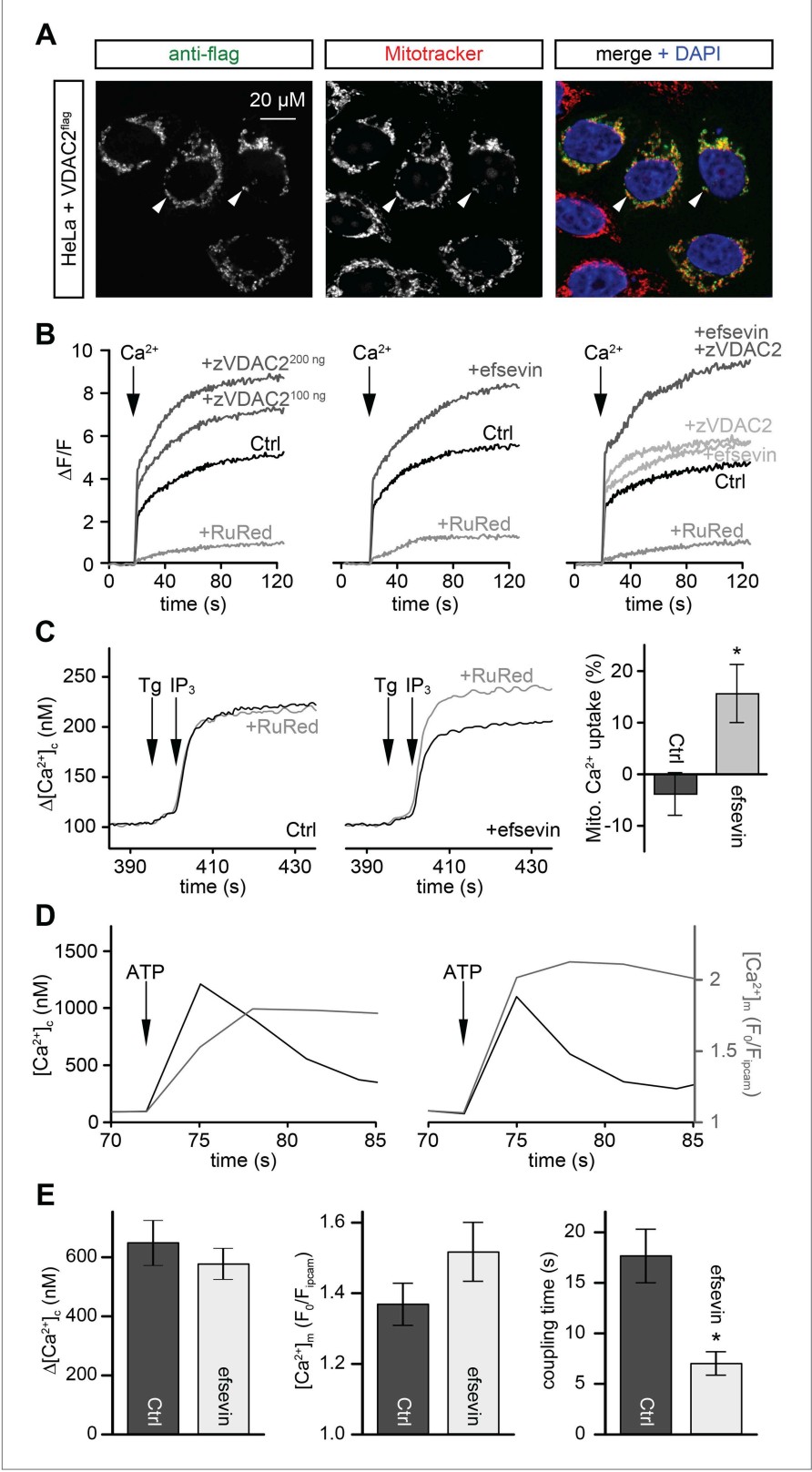

**Figure 5**. Efsevin enhances mitochondrial Ca²⁺ uptake. (**A**) HeLa cells were transfected with a flag-tagged zebrafish VDAC2 (VDAC2$^{flag}$), immunostained against the flag epitope and counterstained for mitochondria with MitoTracker Orange and for nuclei with DAPI. (**B**) Representative traces of mitochondrial matrix [Ca²⁺] ([Ca²⁺]$_m$) detected by
*Figure 5. Continued on next page*

*Figure 5. Continued*

Rhod2. Arrows denote the addition of $Ca^{2+}$. Mitochondrial $Ca^{2+}$ uptake was assessed when VDAC2 was overexpressed (left), cells were treated with 1 µM efsevin (middle) and combination of both at suboptimal doses (right). Control-traces with ruthenium red (RuRed) show mitochondrial specificity of the signal. (**C**) Representative traces of cytosolic $[Ca^{2+}]$ ($[Ca^{2+}]_c$) changes upon the application of 7.5 µM $IP_3$ in the presence (+) or absence (−) of RuRed. Mitochondrial $Ca^{2+}$ uptake was assessed by the difference of the − and + RuRed conditions normalized to the total release (n = 4; mean ± SE). (**D**) MEFs overexpressing zebrafish VDAC2 (polycistronic with mCherry) were stimulated with 1 µM ATP in a nominally $Ca^{2+}$ free buffer. Changes in $[Ca^{2+}]_c$ and $[Ca^{2+}]_m$ were imaged using fura2 and mitochondria-targeted inverse pericam, respectively. Black and gray traces show the $[Ca^{2+}]_c$ (in nM) and $[Ca^{2+}]_m$ ($F_0$/F mtpericam) time courses in the absence (left) or present (right) of efsevin. (**E**) Bar charts: Cell population averages for the peak $[Ca^{2+}]_c$ (left), the corresponding $[Ca^{2+}]_m$ (middle), and the coupling time (time interval between the maximal $[Ca^{2+}]_c$ and $[Ca^{2+}]_m$ responses) in the presence (black, n = 24) or absence (gray, n = 28) of efsevin.

The following figure supplement is available for figure 5:

**Figure supplement 1**. Local $Ca^{2+}$ delivery between IP3 receptors and VDAC2.

healthy looking cells with distinct borders, uniform striations and no membrane blebs or granularity were included in the analysis.

## Biochemistry

For pull down assays mono-N-Boc protected 2,2'-(ethylenedioxy)bis(ethylamine) was attached to the carboxylic ester of efsevin and its derivatives through the amide bond. After removal of the Boc group using TFA, the primary amine was coupled to the carboxylic acid of Affi-Gel 10 Gel (Biorad, Hercules, CA). 2-day-old zebrafish embryos were deyolked by centrifugation before being lysed with Rubinfeld's lysis buffer (*Rubinfeld et al., 1993*). The lysate was precleaned by incubation with Affi-Gel 10 Gel to eliminate non-specific binding. Precleaned lysate was incubated with affinity beads overnight. Proteins were eluted from the affinity beads and separated on SDS-PAGE. Protein bands of interest were excised. Gel plugs were dehydrated in acetonitrile (ACN) and dried completely in a Speedvac. Samples were reduced and alkylated with 10 mM dithiotreitol and 10 mM TCEP solution in 50 mM $NH_4HCO_3$ (30 min at 56°C) and 100 mM iodoacetamide (45 min in dark), respectively. Gel plugs were washed with 50 mM $NH_4HCO_3$, dehydrated with ACN, and dried down in a Speedvac. Gel pieces were then swollen in digestion buffer containing 50 mM $NH_4HCO_3$, and 20.0 ng/µl of chymotrypsin (25°C, overnight). Peptides were extracted with 0.1% TFA in 50% ACN solution, dried down and resuspended in LC buffer A (0.1% formic acid, 2% ACN).

## Mass spectrometry analyses and database searching

Extracted peptides were analyzed by nano-flow LC/MS/MS on a Thermo Orbitrap with dedicated Eksigent nanopump using a reversed phase column (New Objective, Woburn, MA). The flow rate was 200 nl/min for separation: mobile phase A contained 0.1% formic acid, 2% ACN in water, and mobile phase B contained 0.1% formic acid, 20% water in ACN. The gradient used for analyses was linear from 5% B to 50% B over 60 min, then to 95% B over 15 min, and finally keeping constant 95% B for 10 min. Spectra were acquired in data-dependent mode with dynamic exclusion where the instrument selects the top six most abundant ions in the parent spectra for fragmentation. Data were searched against the *Danio rerio* IPI database v3.45 using the SEQUEST algorithm in the BioWorks software program version 3.3.1 SP1. All spectra used for identification had deltaCN>0.1 and met the following Xcorr criteria: >2 (+1), >3 (+2), >4 (+3), and >5 (+4). Searches required full cleavage with the enzyme, ≤4 missed cleavages and were performed with the differential modifications of carbamidomethylation on cysteine and methionine oxidation.

## In situ hybridization

In situ hybridization was performed as previously described (*Chen and Fishman, 1996*). DIG-labeled RNA probe was synthesized using the DIG RNA labeling kit (Roche, Indianapolis, IN).

## Immunostaining

HeLa cells were transfected with a C-terminally flag-tagged zebrafish VDAC1 or VDAC2 in plasmid pCS2+ using Lipofectamine 2000 (Invitrogen). After staining with MitoTracker Orange (Invitrogen)

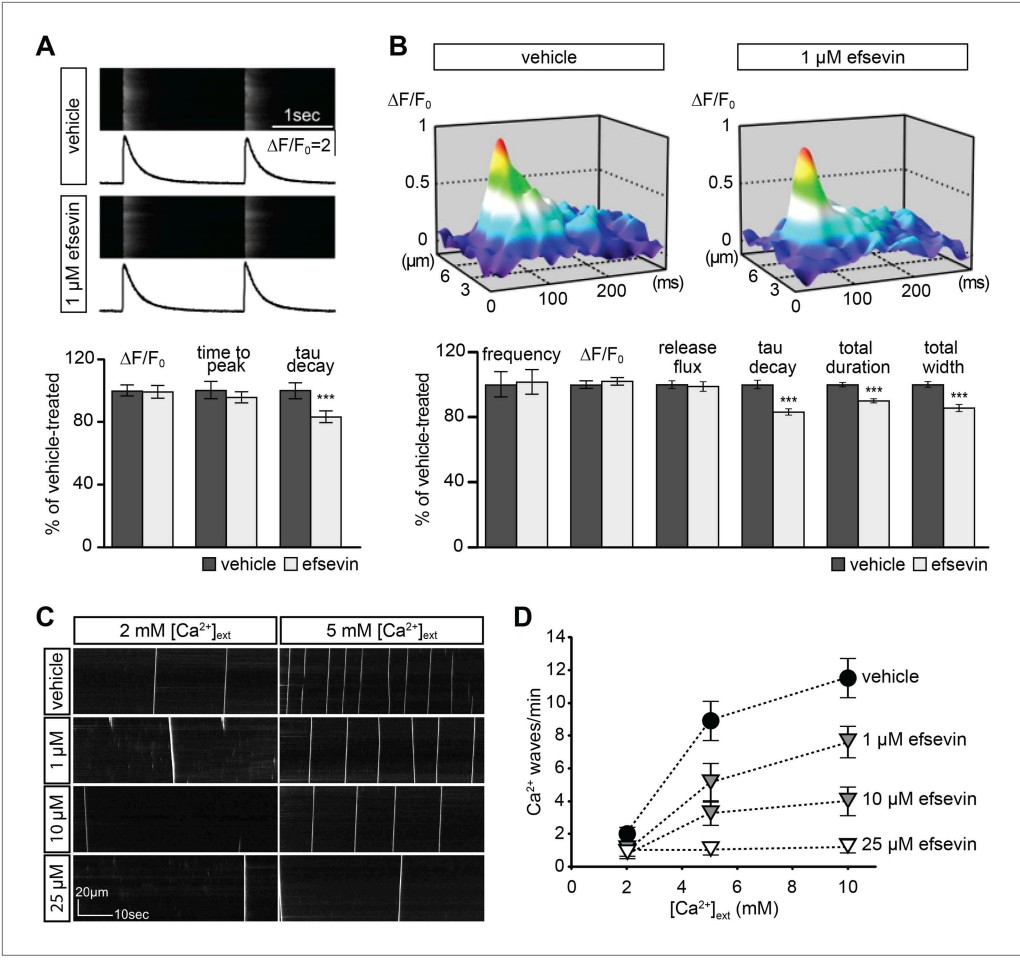

**Figure 6**. Effects of efsevin on isolated cardiomyocytes. (**A**) Electrically paced $Ca^{2+}$ transients at 0.5 Hz (top). Normalized quantification of $Ca^{2+}$ transient parameters reveals no difference for transient amplitude (efsevin-treated at 98.6 ± 4.5% of vehicle-treated) and time to peak (95 ± 3.9%), but a significant decrease for the rate of decay (82.8 ± 4% of vehicle- for efsevin-treated) (lower panel). (**B**) Representation of typical $Ca^{2+}$ sparks of vehicle- and efsevin treated cardiomyocytes (top). No differences were observed for spark frequency (101.1 ± 7.7% for efsevin- compared to vehicle-treated), maximum spark amplitude (101.6 ± 2.5%) and $Ca^{2+}$ release flux (98.7 ± 2.8%). In contrast, the decay phase of the single spark was significantly faster in efsevin treated cells (82.5 ± 2.1% of vehicle-treated). Consequently, total duration of the spark was reduced to 85.7 ± 2% and the total width was reduced to 89.5 ± 1.4% of vehicle-treated cells. *, $p < 0.05$; ***, $p < 0.001$. (**C**) Increasing concentrations of extracellular $Ca^{2+}$ induced a higher frequency of spontaneous propagating $Ca^{2+}$ waves in isolated adult murine ventricular cardiomyocytes. Efsevin treatment reduced $Ca^{2+}$ waves in a dose-dependent manner. (**D**) Quantitative analysis of spontaneous $Ca^{2+}$ waves spanning more than half of the entire cell. Addition of 1 µM efsevin reduced $Ca^{2+}$ waves to approximately half. Increasing the concentration of efsevin to 10 µM further reduced the number of spontaneous $Ca^{2+}$ waves and 25 µM efsevin almost entirely blocked the formation of $Ca^{2+}$ waves.

cells were fixed in 3.7% formaldehyde and permeabilized with acetone. Immunostaining was performed using primary antibody ANTI-FLAG M2 (Sigma Aldrich, St. Luis, MO) at 1:100 and secondary antibody Anti-Mouse IgG1-FITC (Southern Biotechnology Associates, Birmingham, AL) at 1:200. Cells were mounted and counterstained using Vectashield Hard Set with DAPI (Vector Laboratories, UK).

## Mitochondria Ca²⁺ uptake assay in HeLa cells

HeLa cells were transfected with zebrafish VDAC2 using Lipofectamine 2000 (Invitrogen, Carlsbad, CA). 36 hrs after transfection, cells were loaded with 5 µM Rhod2-AM (Invitrogen), a $Ca^{2+}$ indicator preferentially localized in mitochondria, for 1 hr at 15°C followed by a 30 min de-esterification period at 37°C. Subsequently, cells were permeabilized with 100 µM digitonin for 1 min at room temperature.

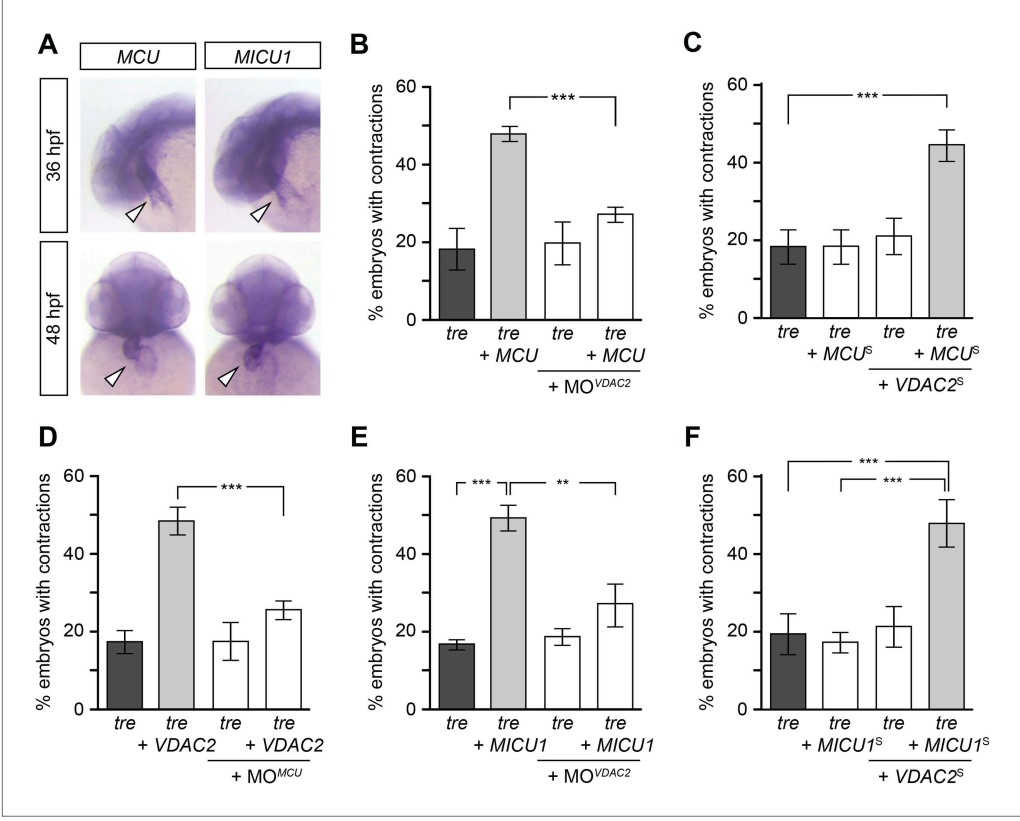

**Figure 7**. Mitochondria regulate cardiac rhythmicity through a VDAC2-dependent mechanism. (**A**) MCU and MICU1 are expressed in the developing zebrafish hearts (arrowhead). (**B**) Overexpression of MCU is sufficient to restore coordinated cardiac contractions in *tre* embryos (47.1 ± 1.6% embryos, n = 112 as opposed to 18.3 ± 5.3% of uninjected siblings, n = 64) while this effect is significantly attenuated when co-injected with morpholino antisense oligonucleotide targeted to VDAC2 (27.1 ± 1.9% embryos, n = 135). (**C**) Suboptimal overexpression of MCU (MCU$^S$) and VDAC2 (VDAC2$^S$) in combination is able to suppress cardiac fibrillation in *tre* embryos (42.9 ± 2.6% embryos, n = 129). (**D**) The ability of VDAC2 to restore rhythmic contractions in *tre* embryos (48.5 ± 3.5% embryos, n = 111) is significantly attenuated when MCU is knocked down by antisense oligonucleotide (MO$^{MCU}$) (25.6 ± 2.4% embryos, n = 115). (**E**) Overexpression of MICU1 is sufficient to restore rhythmic cardiac contractions in *tre* embryos (49.3 ± 3.4% embryos, n = 127 compared to 16.8 ± 1.4% of uninjected siblings, n = 150). This effect is abrogated by VDAC2 knockdown (MO$^{VDAC2}$, 25.3 ± 5.5% embryos, n = 97). (**F**) Suboptimal overexpression of MICU1 (MICU1$^S$) and VDAC2 (VDAC2$^S$) in combination is able to restore rhythmic cardiac contractions in *tre* embryos (48.6 ± 6.0%, n = 106). Error bars represent s.d.; *p < 0.05; ***p < 0.001.

The following figure supplement is available for figure 7:

**Figure supplement 1**. Expression of MCU, MICU1 and VDAC2.

Fluorescence changes in Rhod2 (ex: 544 nm, em: 590 nm) immediately after the addition of Ca$^{2+}$ (final free Ca$^{2+}$ concentration is calculated to be approximately 10 μM using WEBMAXC at http://web.stanford.edu/~cpatton/webmaxcS.htm) were monitored in internal buffer (5 mM K-EGTA, 20 mM HEPES, 100 mM K-aspartate, 40 mM KCl, 1 mM MgCl$_2$, 2 mM maleic acid, 2 mM glutamic acid, 5 mM pyruvic acid, 0.5 mM KH$_2$PO$_4$, 5 mM MgATP, pH adjusted to 7.2 with Trizma base) using a FLUOSTAR plate reader (BMG Labtech, Germany).

## Mitochondria Ca$^{2+}$ uptake assay in VDAC1/VDAC3 double knockout (V1/V3 DKO) MEFs

V1/V3 DKO MEFs were cultured as previously described (*Roy et al., 2009a*). Efsevin-treated (15 μM for 30 min) or mock-treated MEFs were used for measurements of [Ca$^{2+}$]$_c$ in suspensions of permeabilized cells or imaging of [Ca$^{2+}$]$_m$ simultaneously with [Ca$^{2+}$]$_c$ in intact single cells. Permeabilization of the

plasma membrane was performed by digitonin (40 µM/ml). Changes in $[Ca^{2+}]$ in the cytoplasmic buffer upon $IP_3$ (7.5 µM) addition in the presence or absence of ruthenium red (3 µM) was measured by fura2 in a fluorometer (*Csordás et al., 2006*; *Roy et al., 2009b*). To avoid endoplasmic reticulum $Ca^{2+}$ uptake 2 µM thapsigargin was added before $IP_3$. For imaging of $[Ca^{2+}]_m$ and $[Ca^{2+}]_c$, MEFs were co-transfected with plasmids encoding polycistronic zebrafish VDAC2 with mCherry and mitochondria-targeted inverse pericam for 40 hr. Cells were sorted to enrich the transfected cells and attached to glass coverslips. In the final 10 min, of the efsevin or mock-treatment, the cells were also loaded with fura2AM (2.5 µM) and subsequently transferred to the microscope stage. Stimulation with 1 µM ATP was carried out in a nominally $Ca^{2+}$ free buffer. Changes in $[Ca^{2+}]_c$ and $[Ca^{2+}]_m$ were imaged using fura2 (ratio of ex:340 nm–380 nm) and mitochondria-targeted inverse pericam (ex: 495 nm), respectively (*Csordas et al., 2010*).

## Statistics

All values are expressed as mean ± SEM, unless otherwise specified. Significance values are calculated by unpaired student's t-test unless noted otherwise.

## Acknowledgements

The authors thank Kenneth D Philipson, James N Weiss and Adam D Langenbacher for comments on the manuscript, Janice Ahn for assisting the initial chemical screen and Lingling Peng for the synthesis and Yi Chiao Fan for the characterization of efsevin and its derivatives. We also thank Jing Huang, James N Weiss and the UCLA cardiovascular research laboratory for reagents and infrastructure, and Jinghua Tang of UCLA-BSCRC for technical assistance on human ES cell works. We thank William Craigen for providing V1/V3 DKO MEFs.

## Additional information

### Funding

| Funder | Grant reference number | Author |
|---|---|---|
| National Heart, Lung, and Blood Institute | HL081700 and HL096980 | Jau-Nian Chen |
| National Institute of General Medical Sciences | GM071779 and P41GM081282 | Ohyun Kwon |
| The Nakajima Foundation | Graduate Student Fellowship | Hirohito Shimizu |
| China Scholarship Council | Graduate Student Fellowship | Fei Lu |
| University of California, Los Angeles | Philip Whitcome Training Program, Graduate Student Fellowship | Fei Lu |
| Laubisch Foundation | Faculty Award | Jau-Nian Chen |
| Austrian Science Fund | Erwin-Schrodinger Stipendium Postdoctoral Fellowship | Johann Schredelseker |
| University of California, Los Angeles | Broad Stem Cell Research Center Faculty Award | Atsushi Nakano |
| National Heart, Lung, and Blood Institute | HL105699 | Thomas M Vondriska |
| National Heart, Lung, and Blood Institute | HL107674 | Sarah Franklin |
| National Heart, Lung, and Blood Institute | HL070828 | Joshua I Goldhaber |
| National Institute of General Medical Sciences | GM059419 | György Hajnóczky |

The funders had no role in study design, data collection and interpretation, or the decision to submit the work for publication.

### Author contributions

HS, Designed, performed, analyzed, and interpreted experiments, Wrote the manuscript, Conception and design, Acquisition of data, Analysis and interpretation of data, Drafting or revising the article;

JS, Designed, performed, analyzed, and interpreted experiments, Wrote the manuscript, Acquisition of data, Analysis and interpretation of data, Drafting or revising the article; JH, Designed, performed, analyzed, and interpreted experiments, Conception and design, Acquisition of data, Analysis and interpretation of data; KL, Designed and synthesized the compound library and all efsevin used for experiments, Conception and design, Acquisition of data; SN, AE, Examined mitochondrial $Ca^{2+}$ uptake in V1/V3KO MEFs, Acquisition of data, Analysis and interpretation of data; FL, KW, CT, Performed, analyzed, and interpreted experiments, Acquisition of data, Analysis and interpretation of data; SF, Designed and performed the mass-spec analysis, Acquisition of data, Analysis and interpretation of data; HDGF, Designed and synthesized the compound library and all efsevin used for experiments, Acquisition of data, Analysis and interpretation of data; HZ, Designed and performed MEA analysis, Acquisition of data, Analysis and interpretation of data; BL, Performed, analyzed, and interpreted experiments, Acquisition of data; HN, Designed and performed hESC-CM experiments, Acquisition of data, Analysis and interpretation of data; JN, Provided gCaMP construct, Contributed unpublished essential data or reagents; AZS, JKG, Designed and interpreted MEA analysis, Analysis and interpretation of data; AN, Designed and performed hESC-CM experiments, Analysis and interpretation of data; JIG, Supervised physiological analysis, Analysis and interpretation of data; TMV, Designed and performed the mass-spec analysis, Analysis and interpretation of data; GH, Examined mitochondrial $Ca^{2+}$ uptake in V1/V3KO MEFsDesigned and formulated hypothesis, Designed and synthesized the compound library and all efsevin used for experiments, Analysis and interpretation of data; OK, Designed and synthesized the compound library and all efsevin used for experiments, Conception and design, Analysis and interpretation of data; J-NC, Designed and formulated hypothesis, Performed, analyzed, and interpreted experiments, Wrote the manuscript, Conception and design, Acquisition of data, Analysis and interpretation of data, Drafting or revising the article

## Author ORCIDs

Adam Z Stieg, http://orcid.org/0000-0001-7312-9364

## Ethics

Animal experimentation: This study was performed in strict accordance with the recommendations in the Guide for the Care and Use of Laboratory Animals of the National Institutes of Health. All of the animals were handled according to approved institutional animal care and use committee (IACUC) protocols of the University of California, Los Angeles and the Cedars-Sinai Hospital. The protocols were approved by the Cedars-Sinai Institutional Animal Care and Use Committee (#003574 for the use of mouse cardiomyocytes), the Office of Animal Research Oversight that oversees the Ethics of Animal Experiments (ARC# 2000-051-43B for the use of zebrafish) and Embryonic Stem Cell Research Oversight (#2009-006-06 for the use of ES cells) of the University of California, Los Angeles. Every effort was made to minimize suffering.

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
