## [Decision Letter]

Thank you for sending your work entitled “Activation of VDAC2 regulates mitochondrial Ca2+ uptake and cardiac rhythmicity” for consideration at *eLife*. Your article has been favorably evaluated by Vivek Malhotra (Senior editor), a Reviewing editor, and 2 reviewers.

The following individuals responsible for the peer review of your submission have agreed to reveal their identity: Jodi Nunnari (Reviewing editor); Rosario Rizzuto (peer reviewer 1). Reviewer 2 remains anonymous.

The Reviewing editor and the reviewers discussed their comments before we reached this decision, and the Reviewing editor has assembled the following comments to help you prepare a revised submission. All agree that the work is of high quality and represents a significant advance in our understanding of the role of VDAC in calcium handling and cardiac physiology. A majority of the comments pasted below can be addressed by a careful revision of the manuscript. However, as pointed out by reviewer 2, the experiments do not adequately test the authors' conclusion that Ca2+ is directly transferred between the ER and mitochondria. Additional experimentation, for example examining the effect of addition of a Ca2+ chelator.

*Reviewer #1*:

The authors point to mitochondrial Ca2+ buffering as a protective mechanism against erratic electrical activity, and indicates a prominent role of VDAC2 as a “facilitator” of Ca2+ transfer through MCU channels. In this respect, the authors could consider investigating whether additional, possibly isoform-specific mechanisms add to the role of VDAC2 in Ca2+ permeation across the outer mitochondrial membrane.

1) Does VDAC 2 interact with MCU and/or with some of its regulators (MICU1/2/3, EMRE or MCUR1)?

2) Is the expression of MCU and its regulators (“MCU complex”) altered in the tre mutants?

3) Does pharmacological or molecular stimulation of VDAC2 affect the expression of the MCU complex?

*Reviewer #2*:

The manuscript reports exciting results but needs to be improved in the following ways:

1) The results in Figure 3 show a sample of the identification of one VDAC peptide. The mass numbers are not visible and the color code of red and blue is not clear at all especially the blue. The reader is asked to trust the authors and that is not acceptable. If space is a problem then a clear figure should be added to the supplemental file. Further, the authors report the peptides identified that are a match to VDAC 2 but they do not report what peptides were detected that did not match. What fraction of the peptides matched to VDAC2? There is an unfortunate history of attributing action to VDAC that was not due to VDAC and of identifying VDAC as a binding site when another protein was responsible. This manuscript provides other evidence that supports the conclusion that VDAC2 is the target but that does not excuse the failure to honestly report on other proteins present.

2) The authors state “... efsevin restores rhythmic cardiac contraction in tre by potentiating VDAC2 activity.” The authors do not measure VDAC2 activity and thus this conclusion is unnecessary speculation. VDAC2 is not an enzyme and thus its action is not so easily defined in this context. The first part of the last sentence on that page is the appropriate summarizing statement.

3) The authors state: “VDAC is an abundant channel located on the outer mitochondrial membrane serving as a primary passageway for metabolites and ions including Ca2+ (Figure 5) ([44]; 162 [2]; [53]).“ The authors fail to point out that it is the closed state of VDAC that favors Ca++ flux. That seminal publication was not cited. (Biochim Biophys Acta. 2007 Oct;1768(10):2510-5.) The state of VDAC2 as well as its presence is critical.

4) Figure 5 presents some very interesting findings but experimental details are sometimes missing and unclear. In the methods section it talks about perfusion whereas in the figure legend it is called superfusion. It is my understanding that the Fluorostar plate reader does not have perfusion capabilities but rather has the ability to make fluid additions while recording. Is that what was done? Also in the methods section it does not specify the total amount of Ca++ present in the solution, just the presumably calculated free Ca++ concentration. The actual amount of calcium present should be stated and whether the free Ca++ concentration reported is a calculated value of a measured value. Either way it is unclear why the reported value is an exact value. Is this a guess, an approximation?

In addition, the authors state “We examined whether Ca2+ released from intracellular stores could be directly transported into mitochondria through VDAC2 and whether this process could be modulated by efsevin...” The experiments reported do not distinguish between Ca++ release into the medium and quickly taken up my mitochondria or Ca++ directly traveling between the ER and the mitochondria. That could have been tested in 3C by having a strong chelator in the medium. Thus, with the current data the authors cannot claim evidence for direct transfer. They can claim a more effective uptake of Ca++ into mitochondria after efsevin treatment.

Based on the results reported in 5B, the authors conclude that Ca++ has been transported into mitochondria in the permeabilized cells, ignoring the ER. The choice of Rhod2 was presumably based on its affinity for Ca++ and its likely saturation with Ca++ in the ER, so that observed changes are likely due to the mitochondria. The authors make no effort to point this out to the reader. It should be addressed.

5) In the conclusions the authors state: “In this model, VDAC-dependent Ca2+ uptake controls the duration and the diffusion of cytosolic Ca2+ near the Ca2+ release sites...” How, physically, can VDAC control the “duration of cytosolic Ca++” and the “diffusion of cytosolic Ca++”? VDAC cannot control diffusion nor can it control the duration of diffusion. What I hope the authors want to say is that VDAC2 facilitates Ca++ uptake via MCU thus reducing the local Ca++ concentration more rapidly than otherwise.

---

## [Author Response]

We appreciate the reviewers’ insightful and constructive suggestions to help us improve our manuscript. We have taken the reviewers’ advice and revised the manuscript accordingly. In addition to clarifying the text, we revised Figure 3, present new data in Figure 7, and added three supplementary figures. Below, we address each reviewer’s critiques specifically.

*Reviewer #1 questioned whether the expression of MCU complex would be affected by NCX1 mutation or by the molecular stimulation of VDAC2*.

We address this question by in situ hybridization. We find that the expression levels of MCU and MICU1 are comparable between wild type and *tremblor* hearts with and without efsevin treatment. These findings are presented in Figure 7—figure supplement 1.

*Reviewer#1 also questioned whether VDAC2 interacts with MCU and its regulators*.

Our data that restoration of rhythmic contractions in *tremblor* requires the activity of both VDAC2 and MCU provides genetic evidence supporting a functional interaction between VDAC2 and MCU (Figure 7). As the reviewer pointed out, mitochondrial Ca^2+^ transport involves MCU and its regulators. Our model would predict that manipulating the activity of other components of the MCU complex would affect the rescue effect of VDAC2 overexpression. In this revised manuscript, we provide new data showing that: 1) MICU1 is expressed in the developing heart (Figure 7), 2) overexpression of MICU1 restores rhythmic contractions in *tremblor* and this effect requires functional VDAC2 (Figure 7), and 3) there is a synergistic effect of VDAC2 and MICU1 overexpression (Figure 7). These findings provide evidence to support the genetic interaction between VDAC2 and the MCU complex and strengthen our hypothesis that mitochondrial Ca^2+^ buffering is beneficial to maintain cardiac rhythmicity. Future studies on the biochemical interaction between these proteins would provide further mechanistic insights.

*Reviewer #2 made a few suggestions to*
Figure 3:

*The reviewer thought that the sample Mass Spec image presented in*
Figure 3
*was too small for inspection, and questioned the identities of peptides detected in our Mass Spec analysis*.

We thank the reviewer for pointing out this problem. In the original Figure 3, we presented the sample Mass Spec image in the top panel and indicated eight peptides that match VDAC2 in the lower panel. In this revised manuscript, we rearranged Figure 3 as suggested; we present peptide identity in new Figure 3 and provide an enlarged Mass Spec image in Figure 3—figure supplement 1.

*The reviewer also questioned whether we identified any other proteins interacting with efsevin*.

In the course of our study, we inspected six sets of data. VDAC2 was the only protein that was not present in the controls but was consistently identified from the efsevin affinity column. Vitellogenin 1 (vtg1) and NADPH quinine 1 (nqo1) were frequently found in both the control and efsevin affinity columns and thus were considered as contaminants. More importantly, our genetic analysis showed that overexpression of VDAC2 recapitulates the rescue effect of efsevin on *tremblor* and knocking down VDAC2 abolishes the responsiveness of *tremblor* to efsevin. Together, these findings support VDAC2 as a target of efsevin.

*Reviewer #2 raised a few questions regarding the mitochondrial Ca*^*2+*^
*uptake analyses*.

*The reviewer questioned the experimental detail about the mitochondrial Ca*^*2+*^
*uptake experiment in HeLa cells*.

We apologize for the confusing description. In this experiment, we added Ca^2+^ into the samples to make the final Ca^2+^ concentration as approximately 10 μM. We have now clarified this information in the Methods and Figure Legend.

*The reviewer thought it would be helpful to point out how Rhod2 functions as a mitochondrial Ca*^*2+*^
*indicator*.

We appreciate the reviewer’s concern and include this information in the Methods section.

*The reviewer thought that “examine whether Ca*^*2+*^
*released from intracellular stores could be ‘locally’ transported into mitochondria through VDAC2” would be a better description of our analysis*.

We thank the reviewer for the suggestion. We have revised the manuscript accordingly. In addition, we provide new data to support a local IP3R-mitochondrial Ca^2+^ transfer through VDAC2 (Figure 5—figure supplement 1). Previous reports have provided multiple lines of evidence in various cell types to support local Ca^2+^ transfer between IP3 receptors and mitochondria. A commonly used approach is that while both IP3-mediated and SERCA inhibition-induced discharge of the ER Ca^2+^ store cause similar global cytoplasmic [Ca^2+^] signal, only the IP3-induced Ca^2+^ release is rapidly propagated to the mitochondria ([46], Hajnoczky et al. 1995). We now present a similar experiment with V1/V3DKO MEFs to specifically support local Ca^2+^ delivery from IP3 receptors to VDAC2/mitochondria (Figure 5—figure supplement 1).

*Finally, Reviewer #2 made a few editorial suggestions (points 2, 3, 5)*.

We thank the reviewer for the suggestions. We have revised the manuscript accordingly.